

# Theory of kinetically-constrained-models dynamics

**Gianmarco Perrupato[1⋆] and Tommaso Rizzo[2,3]**

**1** Department of Computing Sciences, Bocconi University, 20136 Milano, Italy
**2** Institute of Complex Systems (ISC) - CNR, Rome unit, P.le A. Moro 5, 00185 Rome, Italy
**3** Dipartimento di Fisica, Sapienza Università di Roma, P.le A. Moro 5, 00185 Rome, Italy

⋆ gianmarco.perrupato@unibocconi.it

## Abstract

The mean-field theory of Kinetically-Constrained-Models is developed by considering the Fredrickson-Andersen model on the Bethe lattice. Using certain properties of the dynamics observed in actual numerical experiments we derive asymptotic dynamical equations equal to those of Mode-Coupling-Theory. Analytical predictions obtained for the dynamical exponents are successfully compared with numerical simulations in a wide range of models, including the case of generic values of the connectivity and the facilitation, random pinning and fluctuating facilitation. The theory is thus validated for both continuous and discontinuous transitions and also in the case of higher order critical points characterized by logarithmic decays.

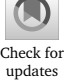

## 1   Introduction

One of the most debated questions in glass physics is whether glassy behavior is caused by a genuine thermodynamic transition that would be observed if one could equilibrate supercooled liquids below the experimental glass transition [1]. Kinetically-Constrained-Models (KCM) [2,3] are often invoked as a proof that such a transition (that is absent in KCMs) is not a logical necessity and that instead dynamic facilitation alone induces the essential features of glassiness, including aging and dynamical heterogeneities, that are well documented numerically in these models. Besides, recent numerical studies [4] performed with the swap technique suggest that dynamic facilitation is indeed at play in supercooled liquids, strengthening earlier insights [5]. Be as it may, it should be noted that the status of KCMs as faithful models of supercooled liquids relies essentially on numerical studies: important advances have been made by the mathematical community [3] but a full theoretical and analytical understanding is still lacking. One important issue is the connection with Mode-Coupling-Theory (MCT) [6] that has been explored by many authors [7–13]. Efforts to describe theoretically KCM dynamics along this line date back to the very first papers on KCMs [14, 15]. In these earlier analytical treatments, approximations were used to derive MCT-like equations, whose solution displays many non-trivial features of the dynamics. However, much as in MCT, they also wrongly predict a *spurious* glass transition that is not at all present in actual systems as studied by numerical simulations, leading many people to dismiss these approaches altogether. Others believe instead that the theory can be fixed and various solutions have been proposed in the literature [16–21] but the issue is still considered open. A recent scenario posits that the approximations involved have a mean-field (MF) nature and it turns out that taking into account fluctuations beyond MF the spurious transition becomes a crossover as observed in realistic systems [22,23]. This opens the possibility that the avoided singularity itself, present in both KCMs, Spin-glasses and supercooled liquids, is the real origin of glassy behavior as observed above the experimental glass transition, independently of the actual mechanism that causes it (*e.g.* facilitation) and also independently of the presence or not of a thermodynamic transition at lower temperatures. At any rate, while MF theory can be worked out analytically in full in the case of fully-connected Spin-Glass models [24,25] and supercooled liquids in the limit of infinite dimensions [26], *a mean-field theory of KCM was still lacking*. In this paper we solve this problem considering KCMs on the Bethe lattices (BL), i.e. finite-connectivity random graphs in which the neighbourhood of a random-chosen site is typically a tree up to a distance that is diverging in the thermodynamic limit. Starting from some simple features of the dynamics as observed in actual numerical experiments we derive exact MCT-like dynamical equations in the most straightforward way. This allows to easily compute the dynamical exponents and the predictions are then successfully verified by extensive numerical simulations in a variety of models.

We consider the Fredrickson-Andersen (FA) KCM [14,15]. Take a system of $N$ independent Ising spins, $s_i \in \{\pm 1\}$, for $i = 1, \ldots, N$, with Hamiltonian $H = \frac{1}{2} \sum_i s_i$. This setup allows for straightforward numerical generation of an initial equilibrium configuration, where the density of spins in the negative state is $p = (1 + e^{-\beta})^{-1}$. Complex behavior occurs because of a dynamic constraint: a spin can flip only if it has at least $f$ (the facilitation) of its $z$ nearest

neighbors in the positive state. A relevant observable is the local persistence $\phi_i(t)$ of the negative (blocking spin). More precisely $\phi_i(t)$ is equal to one at site $i$ if spin $s_i$ is negative for all times $t'$, $0 \le t' \le t$, and zero otherwise. The average persistence $\phi(t) = 1/N \sum_i \phi_i(t)$ counts the fraction of negative spins that never flipped up to time $t$, and it represents an order parameter for the problem. The FA model on the Bethe lattice is known to exhibit dynamical arrest [27–32]: at and below the critical temperature $T_c$ the persistence converges to a plateau value $\phi_{plat}$ that is approached in a power-law fashion, meaning that for $T \le T_c$ typical instances of the system contain an extensive cluster of spins that are blocked at all times. The appearance of such a cluster implies ergodicity breaking. In the ergodicity broken phase, the configuration space divides into an exponential number of equilibrium states [33]. Interestingly, despite the global Boltzmann-Gibbs measure being factorized, conditioning on a state one finds that spins are non-trivially correlated. For a study of the static properties of the equilibrium states in the ergodicity-broken phase, we refer the reader to [33].

The transition happening at $T_c$ is intimately related to bootstrap percolation (BP) (also called $k$-core) [3] because the presence of a BP cluster in the initial configuration implies that the corresponding spins are blocked at all times. As shown in [27] (see also App. A), both $T_c$ and $\phi_{plat}$ can be easily computed by means of this correspondence with BP by the solution of self-consistent equations. In particular for $z = 4$ and $f = 2$ the average persistence $\phi(t)$ obeys at $T_c = 0.480898$ ($p_c = 8/9$):

$$\phi(t) - \phi_{plat} \approx \frac{1}{(t/t_0)^a}, \quad t \gg 1, \tag{1}$$

where $\phi_{plat} = 21/32$.[1] The problem is that through the mapping with BP we can compute the critical temperature and the plateau value (even their fluctuations [34, 35]), but *not* the dynamical exponent $a$. Furthermore numerical simulations [27, 28, 31, 35] have shown that the transition has a Mode-Coupling-Theory nature. This means that for temperatures near $T_c$ there is a $\beta$-regime corresponding to time-scales $\tau_\beta$ on which the persistence is almost equal to $\phi_{plat}$ followed, in the liquid phase ($T > T_c$), by the $\alpha$-regime during which the persistence decays from $\phi_{plat}$ to zero. Within MCT the deviations of the dynamical correlators from the plateau value in the $\beta$ regime is controlled by the following equation [6]:

$$\sigma = -\lambda g^2(t) + \frac{d}{dt} \int_0^t g(t') g(t-t') dt', \tag{2}$$

where $\sigma$ is a linear function of $T_c - T$. In the liquid phase Eq. (2) implies that $g(t)$ leaves the plateau with a $-t^b$ law, and the model-dependent exponents $a$ and $b$ are determined by the so-called parameter exponent $\lambda$ through

$$\lambda = \frac{\Gamma^2(1-a)}{\Gamma(1-2a)} = \frac{\Gamma^2(1+b)}{\Gamma(1+2b)}. \tag{3}$$

From Eq. (2) it also follows that $\tau_\beta$ diverges with $\sigma$ from both sides as $\tau_\beta \propto |\sigma|^{-1/(2a)}$. Similarly the time-scale of the $\alpha$ regime increases as $\tau_\alpha \propto |\sigma|^{-\gamma}$ with $\gamma = 1/(2a) + 1/(2b)$. Sellitto [28] has shown numerically that all the above scaling laws are satisfied in FA models on the BL, as if for some reason the persistence obeyed Eq. (2) with $g(t) \equiv \phi(t) - \phi_{plat}$. In the following we show that this is indeed the case, obtaining also analytical expressions for the exponents $a$ and $b$ through the parameter $\lambda$. We present the argument for the $z = 4$, $f = 2$ case and then extend it to generic values. This allows to demonstrate the theory more

---

[1] The overlap function exhibits similar features to the persistence function, namely in the long-time limit it jumps at $T_c$ from zero to a finite plateau value, which is approached with the same power-law behavior of Eq. (1). See Sec. 4 for a discussion about this point.

broadly, also in presence of continuous transitions, where $\phi_{plat} = 0$. We then examine random pinning, initially investigated numerically in [31] for the FAM, thereby confirming the theory for logarithmic time decays as well. Further validation will come from mixed facilitation models [36].

The paper is organized as follows. In Sec. 2, we derive an exact closed equation of motion for the order parameter, the persistence function, in the $\beta$-regime in the case of FA on the BL with fixed coordination $z = 4$ and facilitation $f = 2$, leaving some technical details to Apps. A and B. In particular, in App. A we discuss the case of generic $z$ and $f$. In Subsec. 3.1 we address the case of FA with continuous transition. In Subsecs. 3.2 and 3.3 we study FA with random pinning and mixed facilitation, respectively. Some details of the random pinning are discussed in Apps. C and E. In Sec. 4 we show that the spin-spin correlation exhibits the same critical behavior as the persistence function. Finally, in Sec. 5 we presents the conclusions. In App. D we provide the details of the numerical simulations.

## 2  Dynamical equations in the $\beta$ regime

To derive the equation, we begin with a number of definitions which are novel to this study. The *blocked persistence* $\phi_b(t)$ is the fraction of negative spins that have been *blocked* at all times less than $t$. Naturally we have $\phi(t) \geq \phi_b(t)$ as a spin that is facilitated (*i.e.* not blocked) does not necessarily flip, however one can argue and confirm numerically (see App. B) that at large times $\delta\phi_b(t) \equiv \phi_b(t) - \phi_{plat}$ and $\delta\phi(t) \equiv \phi(t) - \phi_{plat}$ approach zero with the same leading term $(t/t_0)^{-a}$. More precisely one can argue that the difference $\phi(t) - \phi_b(t)$ is proportional to $d\phi/dt$, and thus it vanishes with a much faster power law $1/t^{a+1}$. We refer the reader to App. B for a complete discussion of this point. We also define the *zero-switch blocked persistence* $\phi_b^{(0)}(t)$ as the fraction of negative sites that have been blocked up to time $t$ because at least three of their neighbors have remained in the negative state at all times less than $t$.

The possible cases can be represented graphically as:

$$\times \quad + \quad \times \quad = \quad \phi_b^{(0)}(t), \tag{4}$$

where the full lines represent the neighbors of the blocked site (circle) which have always remained negative at all times less than $t$, and the dashed line the others.

We also define the *one-switch blocked persistence* $\phi_b^{(1)}(t)$ as the fraction of negative sites that have been blocked because two neighbors have been always negative, a third neighbor has been negative up to some $t'$, and a fourth neighbor has been negative between some time $0 < t'' < t'$ and $t$. Note that this fourth neighbor should not have been negative at all times between 0 and $t$, since this contribution is already counted in $\phi_b^{(0)}(t)$. Again this can be represented graphically:

$$\times \quad = \quad \phi_b^{(1)}(t). \tag{5}$$

The top lines in the diagram (5) represent a *switching couple* of neighbors: the top right line corresponds to a neighbor which is negative up to time $t'$, and the top left line a neighbor which is negative between $t''$ and $t$.

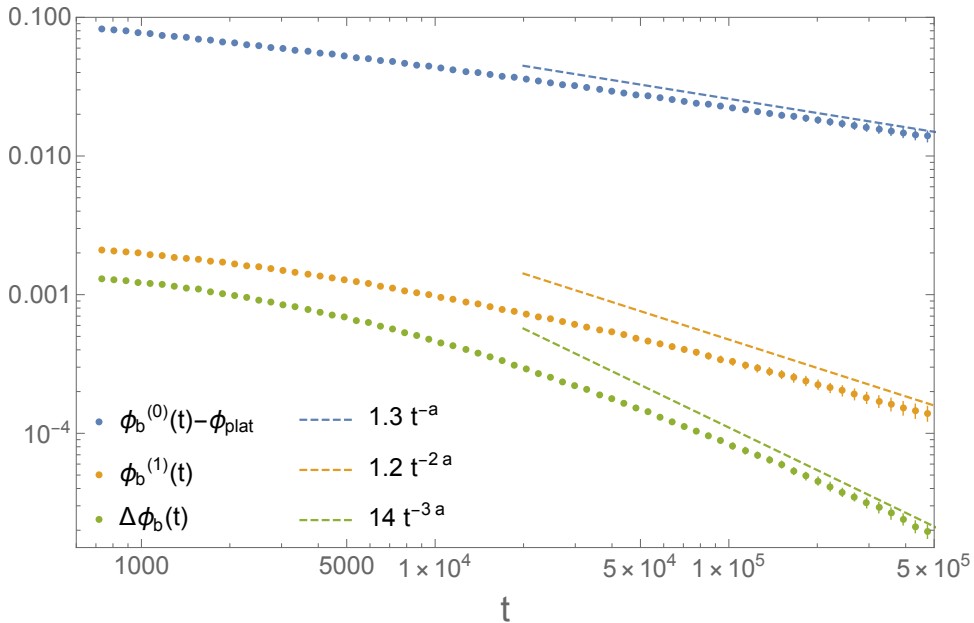

Figure 1: The hierarchy between the different contributions to the blocked persistence as observed in numerical simulations on the BL for $f = 2$, $z = 4$ and $p = p_c$ ($N = 16 \times 10^6$). From top to bottom: $\phi_b^{(0)}(t) - \phi_{plat}$, $\phi_b^{(1)}(t)$, $\Delta \phi_b(t)$. The value of $a$ used to construct the asymptotic (dashed) lines is given by the analytical expression obtained for $\lambda = 2/3$.

To clarify the origins of the names we note that at each time less than $t$ a blocked site has a blocking set, *i.e.* a set of at least three neighbors in the negative (blocking) state. Given a time range $(t, t')$ we say that there is a *minimal blocking set* if the intersection between the blocking sets at all times in the interval is itself a blocking set. Now the *zero-switch* persistence counts those spins for which there is a minimal blocking set in the interval $(0, t)$, while the *one-switch* persistence counts those blocked spins for which there is a minimal blocking set between zero and $t'$ and a different minimal blocking set between $t'$ and $t$. Finally $\Delta \phi_b(t) > 0$ counts all contributions to $\phi_b(t)$ other than $\phi_b^{(0)}(t)$ and $\phi_b^{(1)}(t)$:

$$\phi_b(t) = \phi_b^{(0)}(t) + \phi_b^{(1)}(t) + \Delta \phi_b(t). \tag{6}$$

A crucial observation is that at large times a *hierarchy* between the different contributions emerges, as shown in Fig. 1:

$$1 \gg \phi_b^{(0)}(t) - \phi_{plat} \gg \phi_b^{(1)}(t) \gg \Delta \phi_b(t), \quad t \gg 1. \tag{7}$$

This implies that the critical behavior of $\phi_b(t)$ (and thus of $\phi(t)$ as we said earlier) is given by $\phi_b^{(0)}(t)$ at leading order. In turn, $\phi_b^{(0)}(t)$ can be written exactly in terms of the cavity persistence $\hat{\phi}(t)$, defined as the probability that a site persists in the negative state at all times smaller than $t$ if we force one of its neighbors (the root) to be negative at all times. We have indeed:

$$\phi_b^{(0)}(t) = 4p \, \hat{\phi}(t)^3 (1 - \hat{\phi}(t)) + p \, \hat{\phi}(t)^4. \tag{8}$$

Note that the above equation is the very same that one encounters in bootstrap percolation

$$\phi_{plat} = 4p \, \hat{\phi}_{plat}^3 (1 - \hat{\phi}_{plat}) + p \, \hat{\phi}_{plat}^4, \tag{9}$$

where $\phi_{plat}$ is the probability that a site belongs to the $k$-core, and $\hat{\phi}_{plat}$ is the corresponding cavity quantity. We emphasize that the previous formula holds because one can factorize the contributions from different branches, this is possible due the tree-like structure of the Bethe lattice but would not hold in a generic, say 2D, lattice.

At large times the distance from the plateau value, $\phi_b^{(0)}(t) - \phi_{plat}$, is proportional to the difference between $\hat{\phi}(t)$ and its plateau value $\hat{\phi}_{plat} = 3/4$. In particular at the critical temperature $p_c = 8/9$ we have

$$\phi_b^{(0)}(t) - \phi_{plat} \approx \frac{3}{2} \delta\hat{\phi}(t), \qquad \delta\hat{\phi}(t) \equiv \hat{\phi}(t) - \hat{\phi}_{plat}. \tag{10}$$

The cavity persistence is the typical object that occurs in analytical computations on the Bethe lattice, and indeed in the following we will show that it obeys a self-consistent equation. As we did for the persistence we introduce the blocked cavity persistence $\hat{\phi}_b(t)$, that counts the cavity sites that were blocked at all times $t' < t$. Similarly to the site persistence one can argue that at large times $\hat{\phi}_b(t)$ and $\hat{\phi}(t)$ have the same critical behavior approaching $\hat{\phi}_{plat}$ with the same leading term $(2/3)/(t/t_0)^a$. More precisely one can argue that the difference $\hat{\phi}(t) - \hat{\phi}_b(t)$ is proportional to $d\hat{\phi}/dt$ and thus it vanishes with a much faster power law $1/t^{a+1}$, $\hat{\phi}(t) = \hat{\phi}_b(t) + O(1/t^{a+1})$. The cavity blocked persistence can be also written as a sum of zero-switch and one-switch terms:

$$\hat{\phi}_b(t) = \hat{\phi}_b^{(0)}(t) + \hat{\phi}_b^{(1)}(t) + \Delta\hat{\phi}_b(t), \tag{11}$$

and the crucial hierarchy emerges at large times as well:

$$1 \gg \hat{\phi}_b^{(0)}(t) - \hat{\phi}_{plat} \gg \hat{\phi}_b^{(1)}(t) \gg \Delta\hat{\phi}_b(t), \quad t \gg 1. \tag{12}$$

If we replace $\delta\hat{\phi}_b(t)$ for $\delta\hat{\phi}(t)$ (which is correct at order $O(1/t^{a+1})$) and *neglect* $\Delta\hat{\phi}_b(t)$ (which is correct to order $1/t^{2a}$ according to Fig. 1) we obtain:

$$\delta\hat{\phi}(t) = \delta\hat{\phi}_b^{(0)}(t) + \hat{\phi}_b^{(1)}(t), \tag{13}$$

where both terms in the RHS can be expressed in terms of $\delta\hat{\phi}(t)$ to obtain a closed equation. Let's discuss $\hat{\phi}_b^{(0)}$. The zero-switch cavity persistence is given exactly by:

$$\hat{\phi}_b^{(0)}(t) = 3p\,\hat{\phi}(t)^2(1 - \hat{\phi}(t)) + p\,\hat{\phi}(t)^3, \tag{14}$$

to be compared with the corresponding BP expression:

$$\hat{\phi}_{plat} = 3p\,\hat{\phi}_{plat}^2(1 - \hat{\phi}_{plat}) + p\,\hat{\phi}_{plat}^3. \tag{15}$$

In particular, close to the critical probability $p_c = 8/9$, *i.e.* for small $\delta p \equiv p - p_c$, we have on the time-scale $\tau_\beta$ of the $\beta$-regime:

$$\delta\hat{\phi}_b^{(0)}(t) = \delta\hat{\phi}(t) + \frac{27}{32}\delta p - \frac{4}{3}\delta\hat{\phi}^2(t) + \dots \tag{16}$$

Note that the linear term $\delta\hat{\phi}(t)$ cancels with the LHS of Eq. (13) and thus we have to study the equation at the next order, where $\hat{\phi}_b^{(1)}(t) = O(1/t^{2a})$ contributes. On the other hand at this order it is still correct to neglect $\Delta\hat{\phi}_b(t) = O(1/t^{3a})$.

Let's discuss the second summand of Eq. (13), $\hat{\phi}_b^{(1)}(t)$. According to the definition of $\hat{\phi}_b^{(1)}(t)$ we have one neighbor that remains negative up to a time $t'$, another one that is negative between time $t''$ and $t$ with $0 < t'' < t'$, and a third one that is negative at all times less than $t$.

Table 1: Dynamical parameters of the FA model on the Bethe lattice with connectivity $z$ and facilitation $f$.

| $z$ | $f$ | $p_c$ | $\phi_{plat}$ | $\lambda$ | $a$ | $b$ |
|---|---|---|---|---|---|---|
| 4 | 2 | 0.888889 | 0.65625 | 2/3 | 0.340356 | 0.69661 |
| 5 | 2 | 0.949219 | 0.855967 | 5/8 | 0.355765 | 0.768048 |
| 5 | 3 | 0.724842 | 0.413229 | 0.715095 | 0.32053 | 0.615707 |
| 6 | 2 | 0.970904 | 0.922852 | 3/5 | 0.364399 | 0.812034 |
| 6 | 3 | 0.834884 | 0.657417 | 0.690587 | 0.330849 | 0.656427 |
| 6 | 4 | 0.602788 | 0.294163 | 0.734359 | 0.311953 | 0.583922 |
| 7 | 2 | 0.981146 | 0.95232 | 7/12 | 0.369929 | 0.841922 |
| 7 | 3 | 0.88713 | 0.775028 | 0.672474 | 0.338095 | 0.686806 |
| 7 | 4 | 0.730978 | 0.522658 | 0.719926 | 0.318419 | 0.607721 |
| 7 | 5 | 0.513688 | 0.226228 | 0.744684 | 0.307169 | 0.566936 |

The probability that a cavity site flips between time $t'$ and $t' + dt'$ is given by $-(d\hat{\phi}/dt')dt'$. The total probability that one site is negative between time $t''$ and $t$ with $0 < t'' < t'$ can be computed invoking the reversibility of the dynamics: it is equal to the probability that starting at equilibrium at time $t$, and moving backward in time the site is negative up to time $t - t'$ but not up to time $t$, leading to a factor $\hat{\phi}(t - t') - \hat{\phi}(t)$. As already discussed we have to subtract $\hat{\phi}(t)$ because the case $t' = 0$ (and then $t'' = 0$) leads to a contribution which is already taken into account by the diagram with one dashed leg in Eq. (4). At this point integrating over $t'$, multiplying by a factor six counting all possible switching couples of neighbors, by the probability $p$ of initialising the cavity spin in the negative state, and by the probability $\hat{\phi}(t)$ that the third neighbor remains negative at all times less than $t$ we obtain:

$$\hat{\phi}_b^{(1)}(t) = -6 p \, \hat{\phi}(t) \int_0^t \frac{d\hat{\phi}}{dt'}(t')(\hat{\phi}(t - t') - \hat{\phi}(t)) \, dt'. \tag{17}$$

Note that to write Eq. (17) the local tree-like structure of the Bethe lattice is again crucial, allowing the contributions coming from the unconditioned neighbors of the cavity spin to be considered independent. At this point substituting Eqs. (16) and (17) into Eq. (13), we find that up to second order in $\delta\hat{\phi}(t)$ the cavity persistence satisfies the following closed equation:

$$0 = \frac{27}{32} \delta p - \frac{4}{3} \delta\hat{\phi}^2(t) - 4 \int_0^t \frac{d\hat{\phi}}{dt'}(t')(\hat{\phi}(t - t') - \hat{\phi}(t)) \, dt', \tag{18}$$

where $\delta p = p - p_c$. Integrating by parts, Eq. (18) can be rewritten exactly as the MCT equation (Eq. (2)) with

$$\sigma = \frac{27}{128} \delta p, \qquad \lambda = \frac{2}{3} \rightarrow a = 0.340356. \tag{19}$$

The computation can be extended rather easily to generic $(f, z)$ values. In table 1 we display the results up to $z = 7$ while the complete formula is given in App. A. As we can see from Fig. 2, the predicted values compare well with the numerical data. The small discrepancies can be rationalized recalling that power-laws typically have power-laws corrections, and a more careful procedure is to study the effective exponent $a_{eff} \equiv -d \ln \delta\phi / d \ln t$, that converges to the actual exponent at large times (small values of $\delta\phi$).

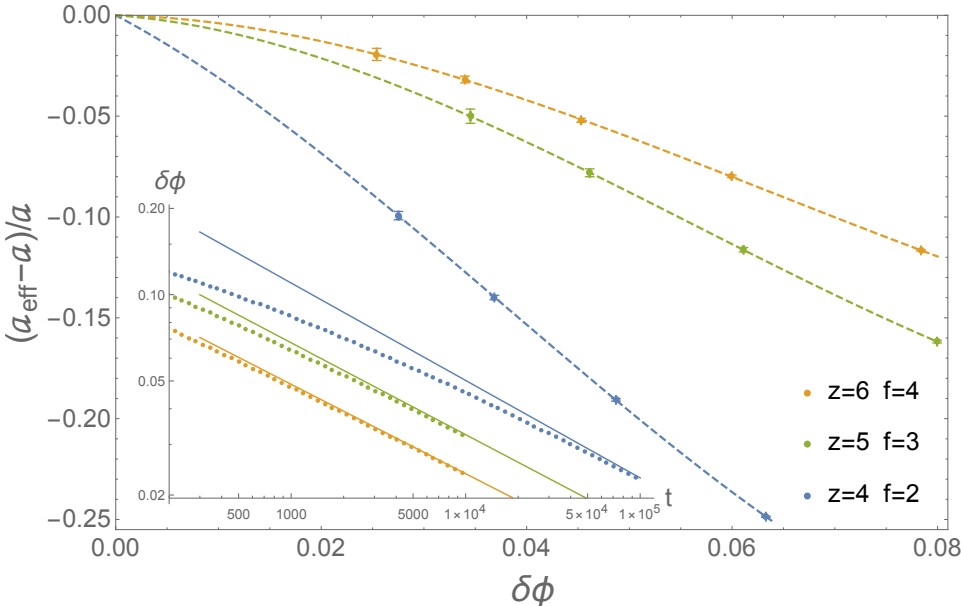

Figure 2: Parametric plot of the relative shift of the effective exponent $a_{eff}$ (see the text) with respect to the analytical prediction $a$ vs. the shift from the plateau. From top to bottom: $z = 6$ $f = 4$, $z = 5$ $f = 3$ and $z = 4$ $f = 2$. Each point is obtained by performing numerical simulations at different sizes ($4 \times 10^6 \leq N \leq 32 \times 10^6$), and then extrapolating to infinite volume. The dashed lines are guides for the eye. Inset: distance of the persistence from the plateau value vs $t$. From bottom to top $z = 6$ $f = 4$, $z = 5$ $f = 3$ and $z = 4$ $f = 2$. The continuous lines correspond to $C_{z,f} t^{-a_{z,f}}$, where the $a_{z,f}$'s are predicted analytically (see Table 1), and $C_{6,4} \approx 0.42$, $C_{5,3} \approx 0.62$ and $C_{4,2} \approx 1.15$.

## 3 Continuous transitions, random pinning and mixed facilitation

### 3.1 Fredrickson-Andersen models with continuous transitions

If $f = z - 1$ the BP transition occurs at $p_c = 1/(z-1)$ and it is continuous, *i.e.* $\phi_{plat}$ is a continuous function of $p$ at $p_c$. This means that $\phi_{plat} = \hat{\phi}_{plat} = 0$ at the transition. One finds that for all values of the connectivity (see App. A), $\hat{\phi}(t)$ decays as $t^{-a}$ with $\lambda = 1/2 \rightarrow a = 0.395263$ for all $z$. However, at variance with the discontinuous case, in which $\delta\phi(t) \propto \delta\hat{\phi}(t)$, $\phi(t)$ is quadratic in $\hat{\phi}(t)$ and thus *its dynamic exponent is doubled*: $\phi(t) \approx z \hat{\phi}^2(t)/2 \propto 1/t^{2a}$. In Fig. 3 we show the persistence for connectivity three, four and five, confirming the prediction that the exponent does not depend on the connectivity.

### 3.2 $A_3$ singularity in random pinning

In [31, 37] Random Pinning (RP) has been considered. RP imposes a further dynamical constraint: once the initial configuration is generated with a given value of $p$, a fraction $c$ of spins drawn at random is not allowed to move. In the $z = 4$, $f = 2$ case, one finds a tricritical point at $c = 1/5$ and $p = 5/6$, where the transition becomes continuous, and the persistence is expected to decay *logarithmically* to a plateau value $\phi_{plat} = 3/8$. The transition is indeed an instance of an $A_3$ singularity [6, 38, 39] that has attracted considerable interest in a number of contexts including attractive liquids [40, 41], confined liquids [42, 43] and randomly pinned liquids [44, 45]. Following the same steps leading to Eq. (18) we find that in this case

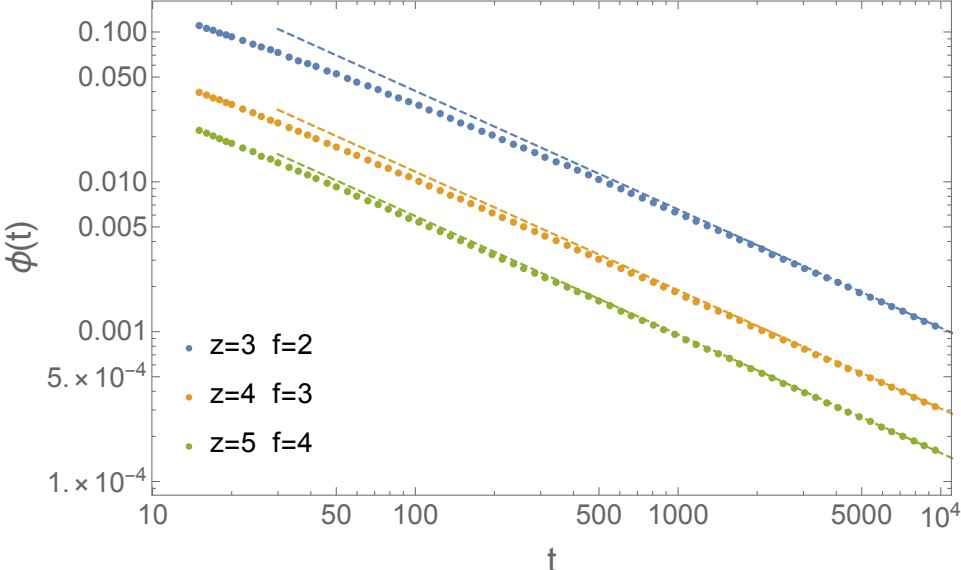

Figure 3: From top to bottom: persistence function $\phi(t)$ for $z = 3, 4, 5$ and $f = z - 1$ (continuous models). The points represent numerical simulations ($N = 16 \times 10^6$). The dashed lines represent the analytical prediction $z/2\,(t/t_0)^{-2a}$. The microscopic time-scale $t_0$, which for $z = 3, 4, 5$ is $t_0 = 1.05, 0.148, 0.0502$, is the unique parameter fitted from the data, while $\delta$ and $a$ are computed analytically (see the text).

(see the App. C) the deviation from $\phi_{plat}$ at the tricritical point is described asymptotically by:

$$0 = \mu\, g^3(t) - g^2(t) + \frac{d}{dt} \int_0^t g(t')\, g(t - t')\, dt', \tag{20}$$

with $\mu = 2/3$, leading to [38]: $g(t) \approx 4\,\zeta(2)\mu^{-1}\ln^{-2}(t/t_0)$ at large times ($\zeta(x)$ is the Riemann Zeta function). In Fig. 4 we plot the effective exponent parametrically, together with i) the leading term, ii) the correction $24\,\zeta(3)\mu^{-1}\ln^{-3}(t/t_0)\ln\ln(t/t_0)$ from Eq. (20) [38] and iii) the solution of a well-known Schematic $F_{12}$ Mode-Coupling-Theory model with parameters tuned to have the predicted asymptotic behavior, see App. E. As expected the effective exponent converges to zero at large times. The parametric expression allows to eliminate the dependence on the unknown timescale $t_0$.

## 3.3 Mixed facilitation models

Models with mixed facilitation display complex phase diagrams also characterized by higher-order singularities [36, 47]. In particular, we considered a $z = 4$ Bethe lattice in which a fraction $c$ of the spins has facilitation three, while the remaining $1 - c$ fraction has facilitation two. In the $(c, p)$ plane there is a line of continuous transitions $p_c = 1/(3c)$ for $c > c_{tric} = 1/2$ where we find $\lambda = 1/(2c)$. In Fig. 5 we display numerical data for the persistence together with the corresponding analytical predictions, again with excellent agreement.

# 4 From the persistence to the correlation

The theory presented so far deals with the time evolution of the persistence $\phi(t)$. As we said before, this is a standard observable in numerical simulations and most importantly allows

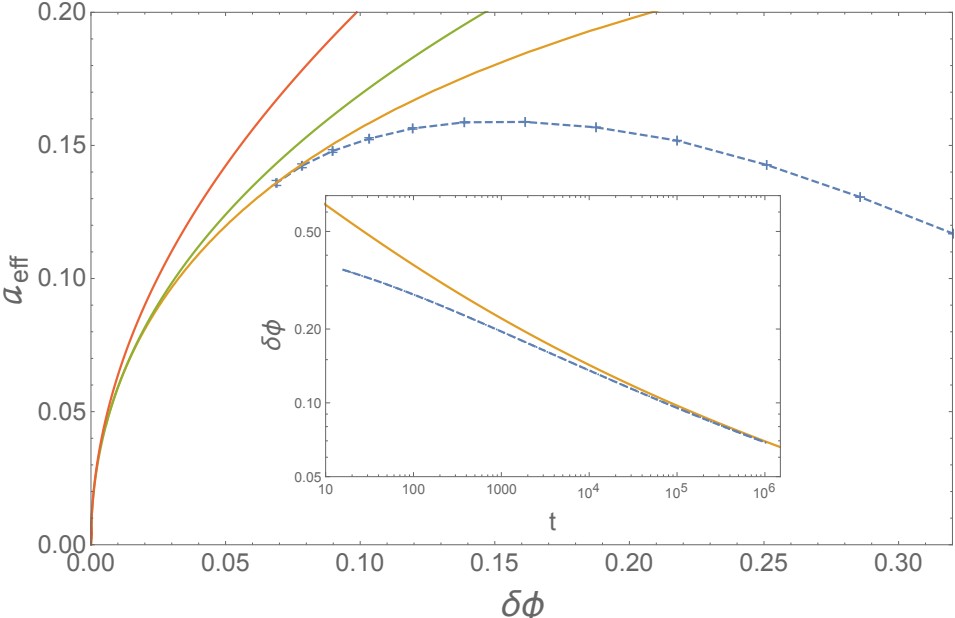

Figure 4: Effective exponent vs. $\delta\phi$ at the $A_3$ singularity of random pinning, see text. Starting from the top left the first two continuous lines are the leading (red) and subleading (green) approximate solutions of Eq. (20). The third line (orange) is the solution of the $F_{12}$ model [46]. The points, interpolated by the dashed line, are numerical data, obtained by averaging over 200 samples of size $N = 16 \times 10^6$. Inset: distance of the persistence from the plateau value $\phi_{plat} = 3/8$ as a function of $t$. Dashed line: numerical data, continuous line: solution of the $F_{12}$ model. The unknown timescale $t_0$ is fitted from the data.

to establish the deep quantitative connection between the FA model and bootstrap percolation. Another important observable, often studied in the literature, is the spin-spin correlation $C(t) \equiv N^{-1} \sum_i^N s_i(0)s_i(t)$. For $(f, z)$ values corresponding to discontinuous transitions, numerical simulations show that $C(t)$ displays the same critical behavior of $\phi(t)$: decreasing the temperature towards $T_c$ it develops a two-step relaxation and below $T_c$ it approaches at infinite times a plateau value $q_{EA}$, in analogy with spin-glass models. We note that while $\phi_{plat}$ can be easily computed by means of the analogy with bootstrap percolation, the overlap $q_{EA}$ obeys more complex iterative equations that we have obtained and solved recently [33]. Thus some questions naturally arise: can we obtain dynamical equations for $C(t)$ as well? Are the dynamical exponents the same? Numerical simulations (see Fig. 6) confirm indeed that this is the case, *i.e.* we have at the critical temperature

$$C(t) - q_{EA} \propto \frac{1}{t^a}, \tag{21}$$

with the same exponent $a$ obtained for the persistence. In the following we will give a simple argument to rationalize this finding.

Below the critical temperature, the ergodicity is broken, and the configuration space of the system divides into an exponential number of equilibrium states, each corresponding to an extensive cluster of spins that are blocked forever (their local magnetization is $m_i = \pm 1$), while the remaining "soft" spins have local magnetization $-1 < m_i < 1$ [33]. An important observation is that, even though the total measure of the problem is factorized, the measure conditioned to one of the equilibrium states is not, since the presence of a blocked cluster induces correlations between the soft spins. This can be visualized by the example in Fig. 7.

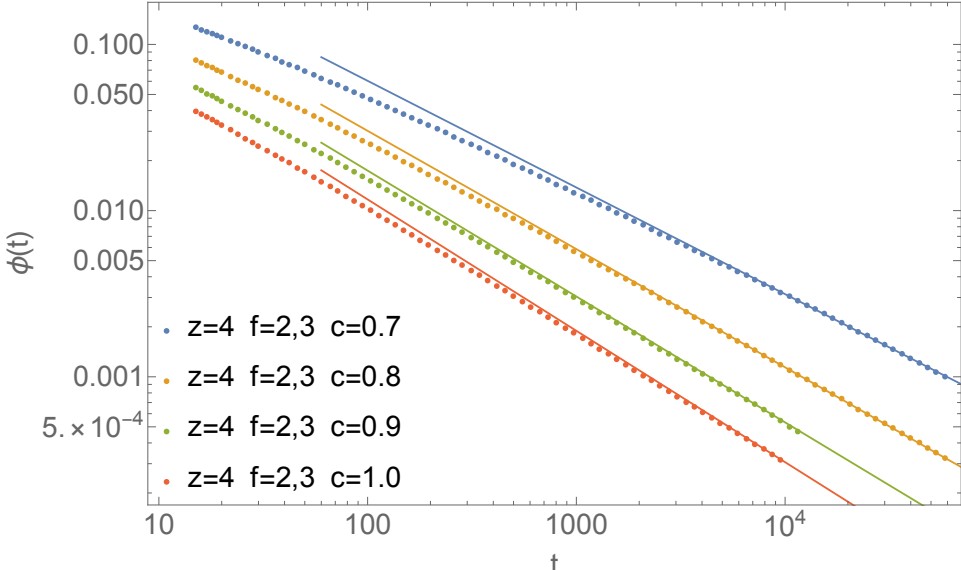

Figure 5: Persistence function $\phi(t)$ of the mixed model $f = 2, 3$ on a Bethe lattice with $z = 4$. From top to bottom $c = 0.7, 0.8, 0.9, 1$ (see the text). The points represent numerical simulations ($N = 16 \times 10^6$). The dashed lines represent the analytical prediction $2(t/t_0)^{-2a}$. In this case $a, t_0$ depend on $c$. The time-scale $t_0$, which for $c = 0.7, 0.8, 0.9, 1$ is $t_0 = 0.44, 0.28, 0.19, 0.145$, is the unique parameter fitted from the data, while $a$ is computed analytically (see the text).

Therefore the magnetization $m_i$ of a soft spin conditioned to one of the equilibrium states is in general different from $(1-2p)$, that is the magnetization computed according to the factorized measure. In analogy with Spin-Glass models one can define the Spin-glass susceptibility

$$\chi_{SG} = \frac{1}{N} \sum_i |\langle s_i s_j \rangle - \langle s_i \rangle \langle s_j \rangle|^2 , \tag{22}$$

that measures the fluctuations of the soft spins inside a given states. Now it turns out that, at variance with spin-glass models, the spin-glass susceptibility remains finite at the critical point. This was observed numerically in [30] and confirmed analytically in [33]. This apparently marginal feature is essential in the following. To make the argument let us sit at $T = T_c^-$ where the blocked cluster has just appeared. As long as $\phi(t)$ has not reached $\phi_{plat}$ there are spins that have not moved yet but will move at later times. Clearly these sites make $C(t)$ different from $q_{EA}$ because their local magnetisation has remained blocked to $\pm 1$ instead of taking its equilibrium value $m_i$. The spins that have moved instead thermalize *rapidly* to the equilibrium value precisely because the soft spins are *not* critical, as implied by the fact that $\chi_{SG}$ remains finite $T = T_c$. In other words the magnetization of a spin that unblocks reaches rapidly its asymptotic value, even if we are at the critical point. It follows that the only thing that determines the deviations of $C(t)$ from $q_{EA}$ is the fact that there is a number of spins that should be soft but have not yet moved and thus *the critical behavior of the overlap is fully controlled by that of the persistence*.

We emphasize again that in order to make the argument it is essential that the fluctuations of the overlap inside a state are not critical and thus as soon as a spin unblocks it quickly reaches equilibrium. In the opposite case we would have seen an additional dynamical slowing down and a slower relaxation of $C(t)$ compared to that of $\phi(t)$.

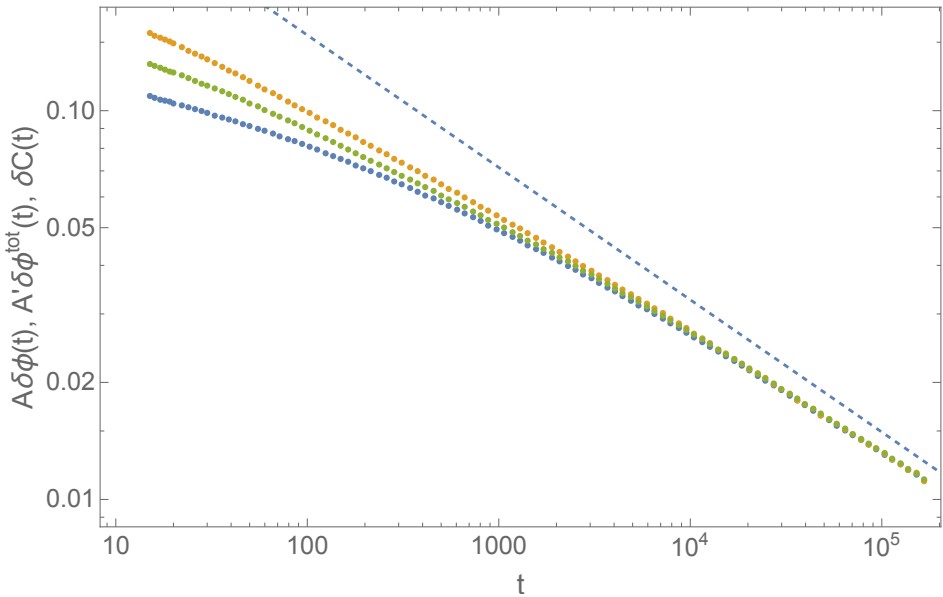

Figure 6: Critical behavior of the correlation in FA with $f = 2$ and $z = 4$ compared with the persistence $\phi(t)$ of the blocked-down spin, and the total persistence $\phi^{tot}(t)$ of all blocked spins. From bottom to top: $A\delta\phi(t)$ (blue dots), $A'\delta\phi^{tot}(t)$ (green dots), $\delta C(t) \equiv C(t) - q_{EA}$ (orange dots), and a reference curve $\propto t^{-a}$ (dashed line), with $a = 0.340356$. The prefactor $A' = 1 - q_{soft}(T_c) \approx 0.548$ is obtained by comparing the square-root behavior of $q_{EA}(T)$ close to $T_c$ (that is computed analytically using the techniques presented in [33]) with that of the plateau value of $\phi^{tot}(T)$ (see Eq. (23)), that can be easily found using the analogy with bootstrap percolation. The prefactor of $\delta\phi(t)$ is $A = A' 143/128 \approx 0.612$. Numerical data are obtained on a system with size $N = 9 \times 10^6$.

The argument extends to temperatures close to $T_c$ either in the liquid or glassy phase, and implies that on the time-scale $\tau_\beta$ of the $\beta$-regime the following relationship holds:

$$C(t) - q_{EA} = A(\phi(t) - \phi_{plat}), \quad T \approx T_c, \quad t = O(\tau_\beta), \tag{23}$$

where $A$ is a constant that depends on $(f, z)$ but *not* on the temperature. Indeed consider the total persistence $\phi^{tot}(t)$, which measures the fraction of spins (both up and down) that have remained unchanged since the initialization. The total persistence has the same critical behavior of $\phi(t)$. In particular, following the arguments of section 2, one finds that:

$$\phi^{tot}(t) - \phi^{tot}_{plat} \approx \frac{143}{128}\left(\phi(t) - \phi_{plat}\right), \quad T \approx T_c, \quad t = O(\tau_\beta), \tag{24}$$

where $\phi^{tot}_{plat} = 2757/4096$ can be easily computed using the analogy with BP. If spins rapidly thermalize after moving for the first time, then for $T \approx T_c$ and $t = O(\tau_\beta)$:

$$C(t) = \phi^{tot}(t) + q_{soft}(T_c)(1 - \phi^{tot}(t)), \tag{25}$$

since the self-overlap of blocked spins is equal to one. In Eq. (25) we introduced the average overlap $q_{soft}(T_c)$ of the soft spins at the critical temperature $T_c$. Therefore, subtracting the plateau values in (25), we find that Eq. (23) holds with $A = A' 143/128$, where $A' = 1 - q_{soft}(T_c)$. Note that $q_{soft}(T)$ is regular at $T_c$, at variance with $q_{EA}(T)$, that has a square-root singularity. The square-root singularity however is only determined by the fact that the fraction of soft spin has a square-root singularity:

$$q_{EA}(T) \approx q_{EA}(T_c) + (1 - q_{soft}(T_c))(\phi^{tot}_{plat}(T) - \phi^{tot}_{plat}(T_c)). \tag{26}$$

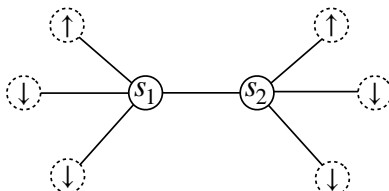

Figure 7: Example of correlations between soft spins in the case $z = 4, f = 2$. Dashed circles represent permanently blocked spins. In order for $s_1$ and $s_2$ to be soft spins, at least one of them should be in the positive state, because if they were both negative they would be permanently blocked.

The quantity $q_{soft}(T_c)$ can be computed by the techniques of [33], comparing the square-root behavior of $q_{EA}(T)$ with that of $\phi_{plat}^{tot}(T)$.

In conclusion we have shown that the critical behavior of $\delta C(t) \equiv C(t) - q_{EA}$ is determined solely by the critical parameter $\delta\phi(t) \equiv \phi(t) - \phi_{plat}$ because $\delta C(t)$ is a linear function of $\delta\phi(t)$ with prefactor $A = A' \, 143/128$, where $A' = 1 - q_{soft}(T_c)$. See Fig. 6 for a comparison between $\delta C(t)$, $\delta\phi(t)$ and $\delta\phi^{tot}(t) \equiv \phi^{tot}(t) - \phi_{plat}^{tot}$.

# 5 Conclusions

We have shown that the persistence of the FA model on the Bethe lattice obeys the critical equation of MCT, *i.e.* Eq. (2). We note that this provides one of the most simple derivations of this equation, being obtained by simple probabilistic arguments. The theory has been extended and validated in a variety of contexts. The possible extension to models with conserved dynamics, notably the Kob-Andersen model [48–50] is left for future work. It is remarkable that the exact asymptotic equation is obtained solely from the assumption that $\Delta\phi_b(t)$ (and $\Delta\hat{\phi}_b(t)$) is negligible at large times according to the hierarchy observed numerically. We are currently investigating the origin of this hierarchy whose understanding should eventually allow to compute systematically the corrections $O(t^{-2a})$, $O(t^{-3a})$, ..., to the leading $t^{-a}$ behavior. Note in particular that from Fig. 1, $\Delta\phi_b(t)$ seems to decay as $t^{-3a}$.

Equation (2) is ubiquitous in glass theory: it has previously been found in the context of supercooled liquids according to MCT [6], mean-field Spin-Glass models with one step of Replica-Symmetry-Breaking [51, 52] and supercooled liquids in the limit of infinite dimensions [26]. We note that, both in spin-glasses and supercooled liquids in infinite dimensions, criticality is associated to the divergence of the static susceptibility inside the glassy states. Instead we have seen in sec. (4) that Eq. (2) holds in KCMs even if the states are not critical, implying that in general criticality in the statics is not a necessary condition for criticality in dynamics.

# Acknowledgments

**Funding information** We acknowledge the financial support of the Simons Foundation (Grant No. 454949, Giorgio Parisi).

# A  The general case $(f, z)$

In this section we derive the closed equation for the persistence, extending the argument presented in Sec. 2 to generic $(f, z)$. The critical probability and the plateau value can be expressed in terms of the function:

$$F(P, k, f_b) \equiv \sum_{i=f_b}^{k} \binom{k}{i} P^i (1-P)^{k-i} \,, \tag{A.1}$$

where we set $k \equiv z - 1$. The parameter $P$ is the cavity probability of the BP cluster, namely the probability that a spin (cavity spin) is blocked if one of its neighbors (the root) is conditioned in the down state, and it obeys the equation:

$$P = p \, F(P, k, f_b) \,, \tag{A.2}$$

where $f_b \equiv k + 1 - f$ is the number of neighbors that must be blocked in the negative state (besides the root) for the cavity spin to be blocked in the negative state. At the critical temperature the above equation develops a solution with $P \neq 0$. In the discontinuous case $P$ jumps from zero to a finite value at $P_c$. The finite value can be determined by the equation

$$\left( F(P_c, k, f_b) - P_c \left. \frac{dF(P, k, f_b)}{dP} \right|_{P=P_c} \right) P_c^{-f_b} = 0 \,. \tag{A.3}$$

Note that the above equation is a polynomial of degree $k - f_b$, and thus it is linear for $f = 2$, and quadratic for $f = 3$. The critical probability is given by:

$$p_c = P_c / F(P_c, k, f_b) \,, \tag{A.4}$$

while the plateau value is given by:

$$\phi_{plat} = p_c \, F(P_c, k+1, f_b+1) \,. \tag{A.5}$$

For $f = z - 1$ we have $f_b = 1$, and the lowest power of $P$ in the function $F$ is one, implying a continuous transition ($\phi_{plat} = \hat{\phi}_{plat} = 0$) with

$$p_c = 1 \Big/ \left( \left. \frac{dF(P, k, 1)}{dP} \right|_{P=0} \right) = \frac{1}{k} \,. \tag{A.6}$$

At this point, in order to compute the dynamical equation we have to study $\hat{\phi}_b^{(1)}(t)$ (see the main text) at the second order in $\delta\hat{\phi}$. Consider the cavity spin. We are interested in the case in which: $f_b - 2$ neighbors (beside the root) are always blocked down up to time $t$, one neighbor is blocked down from 0 to $t' < t$, another neighbor is blocked down from $t'' < t'$ to $t$. Following the same arguments of the case $(4, 2)$, at the second order in $\delta\hat{\phi}$ we have:

$$\hat{\phi}_b^{(1)}(t) = p \, C_{k, f_b} \, P_c^{f_b - 1} (1 - P_c)^{k - f_b - 1} \int_0^t \left( -\frac{d\hat{\phi}}{dt'}(t') \right) \left( \hat{\phi}(t - t') - \hat{\phi}(t) \right) dt' \,, \tag{A.7}$$

where the combinatorial factor

$$C_{k, f_b} = \binom{k}{f_b - 1} (k - f_b + 1)(k - f_b) \,, \tag{A.8}$$

counts all possible couples of neighbours such that one of them is blocked down from 0 to $t' < t$, and the other is blocked down from $t'' < t'$ to $t$. Thus the closed equation for $\delta\hat{\phi}(t)$ becomes:

$$0 = F(P_c, k, f_b)\delta p + p\frac{1}{2}\frac{d^2 F(P, k, f_b)}{dP^2}\bigg|_{P=P_c}\delta\hat{\phi}^2(t)$$
$$+ p\, C_{k,f_b}\, P_c^{f_b-1}(1-P_c)^{k-f_b-1}\int_0^t\left(-\frac{d\hat{\phi}}{dt'}(t')\right)\left(\hat{\phi}(t-t')-\hat{\phi}(t)\right)dt'. \qquad (A.9)$$

At this point integrating by part, we can write Eq. (A.9) in the MCT form [6]:

$$\sigma = -\lambda\,\delta\hat{\phi}^2(t) + \frac{d}{dt}\int_0^t\delta\hat{\phi}(t')\,\delta\hat{\phi}(t-t')\,dt', \qquad (A.10)$$

finding the two parameters $\sigma$ and $\lambda$:

$$\sigma = \frac{F(P_c, k, f_b)}{p_c\, C_{k,f_b}\, P_c^{f_b-1}(1-P_c)^{k-f_b-1}}\,\delta p, \qquad (A.11)$$

$$\lambda = 1 + \frac{\frac{1}{2}\frac{d^2 F(P, k, f_b)}{dP^2}\bigg|_{P=P_c}}{C_{k,f_b}\, P_c^{f_b-1}(1-P_c)^{k-f_b-1}}. \qquad (A.12)$$

In particular for $f = 2$ we have $\lambda = \frac{1+k}{2k}$. In the continuous case Eqs. (A.11) and (A.12) becomes

$$\sigma = \frac{1}{k-1}\delta p, \qquad \lambda = \frac{1}{2}, \qquad (A.13)$$

however, as already discussed, at variance with the discontinuous case, $\phi(t)$ is quadratic in $\hat{\phi}(t)$ and thus its dynamic exponent is doubled: $\phi(t) \approx z\,\hat{\phi}^2(t)/2 \propto 1/t^{2a}$.

## B Difference between the persistence and the blocked persistence

As discussed in the main text the persistence $\phi(t)$, the blocked persistence $\phi_b(t)$, and the zero-switch persistence $\phi^{(0)}(t)$ all have the *same* critical behavior. This is easily observed in numerical simulations (see Fig. 9). In this section we want to discuss an argument for justifying this result. Let us note that in principle a spin could have been facilitated at some time in the past but did not switch due to a thermal fluctuation. However it is clear that the higher the number of times that it was facilitated, the lower the probability that it did not switch. Now due to the reversible nature of the dynamics, if the spin was facilitated at some distant time $t'$ in the past with probability one, it must have been facilitated many times at later times, leading to a vanishing probability that it did not switch. In other words we expect that once a site becomes facilitated, it will switch with probability one after a finite time $t_{sw}$ that is short on the time scale of the critical dynamics. The only possibility is that the site has become facilitated at a time $t'$ close to $t$, *i.e.* $t - t' = O(t_{sw})$. On the other hand the number of sites that become facilitated between times $t - t_{sw}$ and $t$ is given by

$$\phi_b(t - t_{sw}) - \phi_b(t) \approx -t_{sw}\frac{d\phi_b(t)}{dt} \ll \phi_b(t), \qquad (B.1)$$

thus we expect that the difference between $\phi(t)$ and $\phi_b(t)$ is proportional $O(1/t^{a+1})$ at large times, and that it can be neglected with respect to $1/t^a$. The argument is confirmed by the numerical data (see Fig. 8).

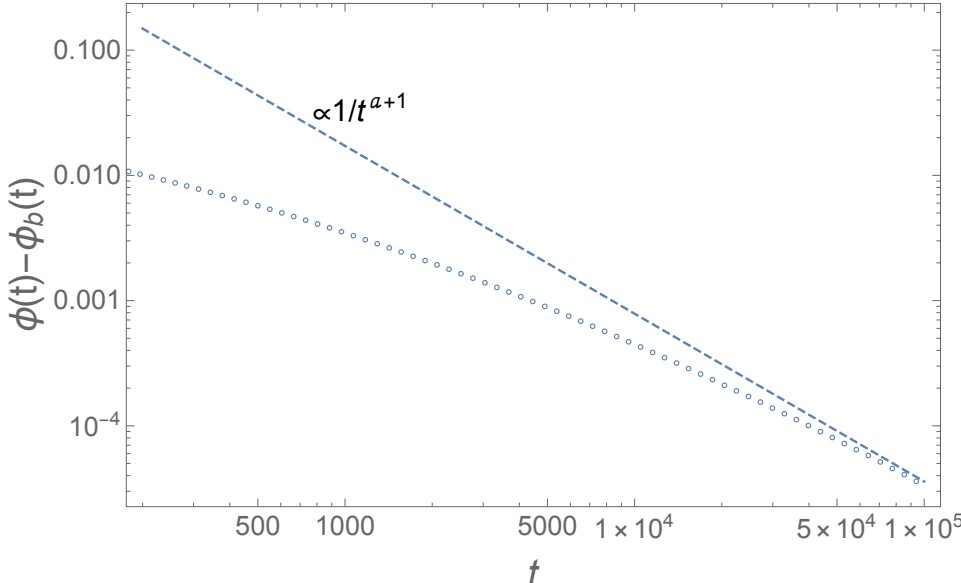

Figure 8: Difference between the average local persistence $\phi(t)$ and $\phi_b(t)$ (the average local persistence of the sites that have never been facilitated up to time $t$) in the case of $z = 4$ and $f = 2$ at the critical temperature. The dashed line is the expectation $C/t^{a+1} \propto d\phi/dt$, $C \approx 180$. The data correspond to the average of 80 samples of size $N = 16 \times 10^6$.

## C  Random pinning

In the random pinning variation of the Fredrickson-Andersen model, after drawing the initial condition, a fraction $c$ of sites selected at random are frozen (pinned), i.e. they are not updated through the dynamics. In the case $z = 4, f = 2$, that we studied in the main text, the cavity probability of being blocked down is given by:

$$P = p\,c + p\,(1-c)\,F(P,3,2),\tag{C.1}$$

where the function $F$ is defined in Eq. (A.1). In the $c-p$ plane Eq. (C.1) determines a critical line (see Fig. 10), which can be computed solving the following system of equations:

$$
\begin{aligned}
1 &= p\,(1-c)\,\frac{dF(P,3,2)}{dP} = 6\,(1-c)\,p\,P\,(1-P),\\
0 &= F(P,3,2) - P\,\frac{dF(P,3,2)}{dP} + \frac{c}{1-c} = P^2\,(4P-3) + \frac{c}{1-c}\,.
\end{aligned}
\tag{C.2}
$$

The plateau value of the persistence $\phi_{plat}$ is connected to the value of $P$ through

$$\phi_{plat} = p\,c + p\,(1-c)\,F(P,4,3).\tag{C.3}$$

For $0 \le c < 1/5$, when $p$ is small, Eq. (C.1) admits a solution because of the fraction of pinned spins, and of a small fraction of unpinned spins which are blocked due to the presence of neighboring spins which are pinned down.

By increasing $p$ one finds another solution which appears discontinuously at the transition line. From a dynamical point of view this singularity is analogous to that obtained at $c = 0$. In particular the expression for the $\lambda$ parameter exponent is given by expression Eq. (A.12), where in this case the critical cavity probability $P_c$ depends on the fraction $c$ of pinned spins through Eq. (C.2).

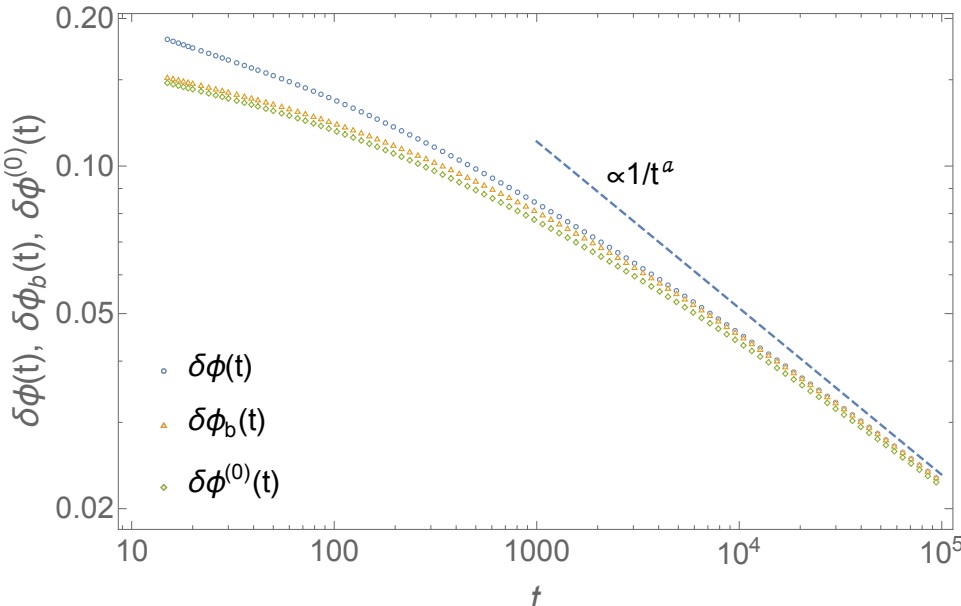

Figure 9: From bottom to top: zero-blocked persistence $\delta\phi^0(t) = \phi^0(t) - \phi_{plat}$, blocked persistence $\delta\phi_b(t) = \phi_b(t) - \phi_{plat}$, and persistence $\delta\phi(t) = \phi^0(t) - \phi_{plat}$ for $z = 4$ and $f = 2$ at the critical point. In this case $p_c = 8/9$, $\phi_{plat} = 21/32$ and $a = 0.340356$. The data correspond to averages over 80 samples of size $N = 16 \times 10^6$.

Increasing $c$, the jump of $\phi_{plat}$ at the critical line gets smaller and smaller and it vanishes for $c = 1/5, p = 5/6$, where the transition becomes continuous. This point is found by adding to system (C.2) the condition

$$0 = \frac{d^2 F(P,3,2)}{dP^2},\tag{C.4}$$

which implies that in the equation for the dynamics, instead of quadratic term $\delta\phi^2$ (see Eq. (A.9)) here there is a cube $\delta\phi^3$. Indeed at the continuous critical point one finds:

$$0 = \frac{1}{6}\frac{d^3 F(P,3,2)}{dP^3}\bigg|_{P=P_c} \delta\hat\phi^3(t) + 6P_c \int_0^t \left(-\frac{d\hat\phi}{dt'}(t')\right)\left(\hat\phi(t-t') - \hat\phi(t)\right) dt'.\tag{C.5}$$

Equation (C.5) corresponds in the MCT language to an $A_3$ singularity which, as discussed in the main text, is associated with a logarithmic decay of the persistence.

## D  Numerical simulations

The numerical simulations have been performed according to the following scheme. The first step is the generation of the graph. In our case we consider a Bethe lattice with fixed coordination $z$. More precisely we start from an "elementary cell" $\mathcal{C}$ with $n$ nodes, such that each node has $z$ neighbors. After that we create $M$ replicas, $\mathcal{C}^1,\ldots,\mathcal{C}^M$ of the cell. In this way each site $i$ has $M$ replicas, that we denote by $\sigma_i^a$, where $a = 1,\ldots,M$ is the replica index. At this point we define a new graph. For each edge $(i,j)$ of the cell, we generate a random permutation $\mathcal{P}$ of $(1,2,\ldots,M)$, and, for each $a$, we replace the edge connecting $\sigma_i^a$ to $\sigma_j^a$ with an edge connecting $\sigma_i^a$ to $\sigma_j^{\mathcal{P}(a)}$. Note that this procedure, the so-called $M$-layer construction [53], does not change the coordination of the nodes. In this way, as shown in [53], one obtains for large $M$ an instance of Bethe lattice (the density of cycles of fixed length is vanishing for

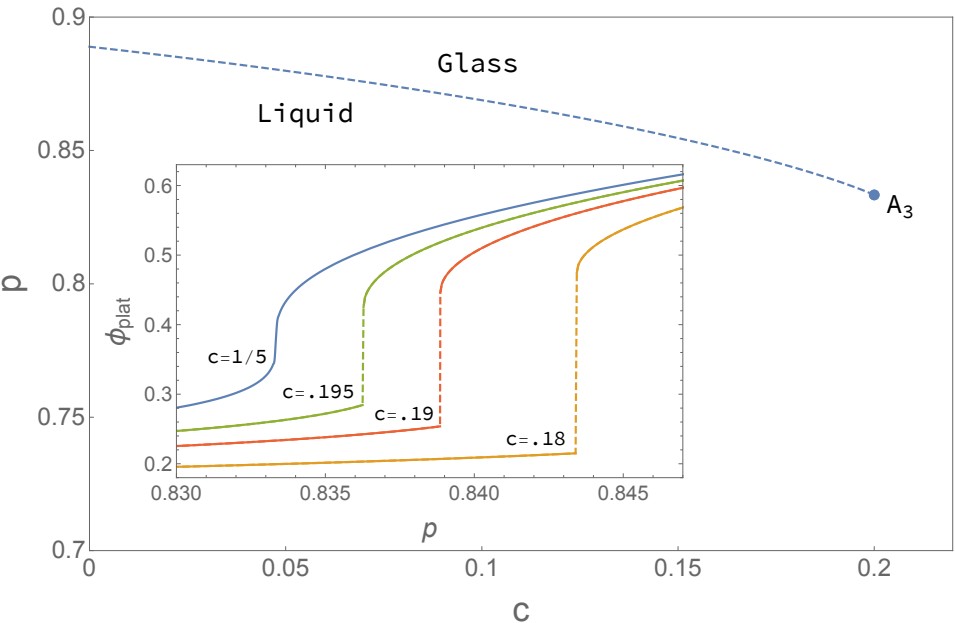

Figure 10: Phase diagram of the random pinning with $z = 4, f = 2$. The dashed line corresponds to a discontinuous transition ($A_2$ transition in the MCT terminology) line. At the terminal point ($c = 1/5, p = 5/6$) the transition becomes continuous with a logarithmic decay of the persistence ($A_3$ MCT transition). In the inset: $\phi_{plat}$ as a function of $p$ at fixed value of $c$. From right to left the first three curves correspond to $c = 0.18, 0.19, 195$, and the last curve is obtained at $c = 1/5$, crossing the $A_3$ point. Note that at variance with the unpinned case here $\phi_{plat}$ is in general different from zero also in the liquid phase.

$M \to \infty$). The simulations discussed in the text are performed on lattices with coordination $z = 3, 4, 5, 6$. The cases $z = 4, 6$ are obtained starting from, respectively, a square and a cubic cell. The cells for the cases $z = 3, 5$ are shown in Fig. 11. In all cases we start from elementary cells which are bipartite, i.e. each node can be associated with either, say, a "black" or "white" label, in such a way as two nodes of the same color are not connected. As we will discuss shortly this is a particularly convenient choice for the dynamics. It is worth noticing that the $M$-layer construction conserves the bipartition property of the cells.

The second step is the generation of the initial configuration. This part is trivial in KCMs since the probability distribution of the initial configuration is factorized on the sites of the lattice. After these two steps we are given an instance of the problem, that we want to evolve with the dynamics. We mainly used Metropolis moves (a negative mobile spin is flipped with probability $e^{-\beta}$ and a positive mobile spin is flipped with probability one) with a chessboard updating sequence (all black spins are updated sequentially and then all the white spins are updated sequentially). A fundamental observation [35] is that other dynamics (e.g. Glauber) and updating orders (e.g. random order) at large enough times produce curves which differ only by a constant shift in time, that in the mode-coupling equation affects only the unknown time-scale constant $t_0$. As already observed in [35], the chessboard/Metropolis scheme turns out to be the most convenient in terms of CPU time and relaxation time of the dynamics.

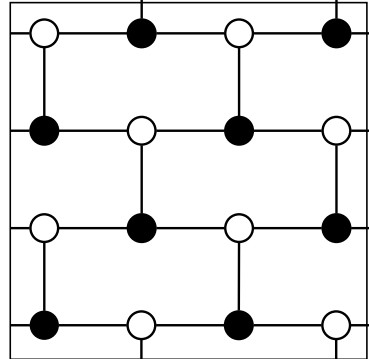 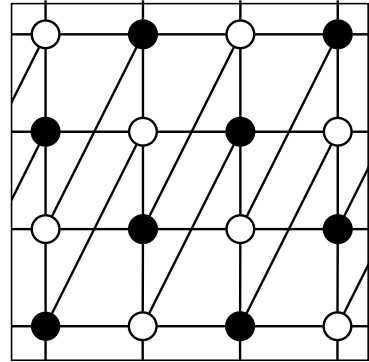

Figure 11: Example of "elementary cells". On the left the case $z = 3$, on the right $z = 5$. The cells have periodic boundary conditions.

## E  The $F_{12}$ model

The data shown in Fig. 4 of the main text were obtained solving numerically the following equation:

$$\dot{g}(t) + g(t) + \int_0^t d\tau\, K(t-\tau)\dot{g}(\tau) = 0\,, \tag{E.1}$$

with

$$K(t) = g(t) + g^2(t)\,, \quad g(0) = 1\,. \tag{E.2}$$

The asymptotic behavior of the previous equation corresponds to the asymptotic behavior of equation (10) in the main text with $\mu = 1$. To obtain a solution corresponding to generic $\mu$ one has to divide the solution of (E.1) by $\mu$. The data shown in the main text are obtained using the gitHub library [46].

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
