# Peer review of "Theory of Kinetically-Constrained-Models Dynamics"

_SciPost Physics, doi:SciPost Phys. 18, 020 (2025)_

## Round 1 · Referee Report · Anonymous (Referee 1) · 2024-10-14

Strengths

  1. Studies the dynamics of a class of kinetically constrained models (f-spin facilitated Fredrikson-Andersen models, part of a family of KCMs which are very relevant for understanding glasses).
  2. By placing the models in a z-tree (rather than on a lattice with loops) the problem becomes effectively mean field, and therefore amenable to a technique like mode-coupling theory (and specifically schematic MCT in this case).
  3. The paper obtains the MCT exponents exactly, thus completing the MCT study of such Bethe-lattice KCMs.
  4. The main result is that of eq.18/19, with table 1 giving the exponent as a function of f and z.

Weaknesses

The paper delivers what it purports to do, so in that sense I cannot see any weaknesses. It is written in a way that will be accessible to experts in this field (especially those like me that have been around it since the time MCT was more fashionable).

Report

This paper studies the dynamics of a class of kinetically constrained models (f-spin facilitated Fredrikson-Andersen models). These are part of a family of KCMs which are very relevant for understanding glasses (cf. East model or constrained lattice gases). In an f-FA model a spin can flip if f of its neighbours are in a specific state (say up), this kinetic constraint leading to slow relaxation when that state is scarce (e.g. at low T when there are few of the energetically expensive ups). KCMs in low dimensional square lattices, and with low f (like the FA or East models) behave in a very non mean-field way. Interestingly, when place in mean-field like geometries, such as a Cayley tree, they can display mean-field like dynamics, in particular for larger f (when the KCM is more constrained) when the KCM has an ergodicity breaking singularity as some finite average density of facilitating sites (i.e. finite temperatures).

Early work had shown schematic MCT like behaviour of KCMs in trees, but this programme of work was not completed. This is what this paper mostly does. Specifically, by considering the persistence function (which is an alternative to the autocorrelation function to quantify relaxation, and often simpler to study) the paper finds the MCT exponents of f-FA in z-trees exactly, thus completing the MCT study of such Bethe-lattice KCMs. The main result is that of eq.18/19, with table 1 giving the exponent as a function of f and z. There are more results, like doing the same in the case of random pinning (which is known in other mean-field systems to lead to a higher order MCT singularity), and they also argue on convincing grounds that their results for the persistence also capture the behavior of the autocorrelator in this kind of geometries.

The paper is easy to follow for those familiar with MCT methods. The main text deals with the essential aspects of the approach that leads to the central results, with other technical details relegated to the appendices. Overall this is a solid piece of work that fills gaps in the mean-field study of KCMs. I recommend publication as is.

Recommendation

Publish (meets expectations and criteria for this Journal)

---

## Round 1 · Referee Report · Anonymous (Referee 2) · 2024-11-1

Strengths

This is an interesting paper about the spin dynamics of Fredricksen-Andersen (FA) models on Bethe lattices. The persistence function is shown to obey an equation consistent with mode-coupling theory (MCT) and the implications of this equation are studied, including connection to existing results from bootstrap percolation. The results are verified by numerical simulations.

Weaknesses

Some aspects of the presentation could be clearer.

Report

Given the strengths noted above, these results deserve publication in scipost. However, the authors should consider the following points.

  • Some of the previous works for FA models on Bethe lattices (eg [27-32]) should be cited in the introduction (instead of later in Sec 2). Their relation to the current work should be explained in more detail, to give context.

  • Some of the results of this work are quoted in Ref[34] which is now published as Phys Rev E 110, 044312 (2024). In that work they are attributed to arxiv:2212.05132, which seems to be a precursor of this work, albeit with a different title. I would suggest to include a comment in the introduction to explain the relationship between Ref[34] and this work.

  • In the first paragraph of section 2.1, the authors use several terminologies for the state of spin i. Either negative/positive (presumably corresponding to $s_i=\mp1$) or up/down or excited/unexcited. It would be useful to introduce a single terminology from the start, and stick to it.

  • I am not sure if the" zero-switch"/"one-switch" sets defined above eq(6) are new in this work, or taken from the literature. A comment would be helpful.

  • Section 2.6 draws heavily on Ref 34, which is itself quite long and technical. The explanations that appear between equations (21) and (22) of this work are interesting but not easy to follow. For example:

. I am not sure what means an "equilibrium state" in this context. The authors may want to provide a precise definition, or perhaps just a description in words. For a fixed instance of the Bethe lattice then it seems to me that there are natural "states" such that : each state has a set of frozen spins, and all other spins s_i are negative with probability p and positive with probability 1-p. (These would correspond to communication classes within the (reducible) Markov chain that describes the system). However, I am not sure if the authors refer to these states, or some other ones.

. Please also define the $m_i$. In the example "states" above then all the soft spins would have $m_i = 1-2p$, and the spin correlations also factorise. It seems that the authors have found a more complicated situation than this, this deserves a (short) physical explanation so that the paper makes sense without requiring the reader to read all of Ref[34] in detail.

  • It would be useful in the conclusion to briefly return to the question raised in the introduction about the connection of static and dynamic quantities. Do the results of this work offer any new perspectives on this question? For example, my understanding of Sec 2.6, is that this model has a close connection between statics and dynamics, but on the other hand the spin-glass susceptibility is regular at dynamical arrest. Could this scenario be generic in some larger class of models, and can be attributed to some underlying physical/mathematical structure?

  • In appendices I would suggest to write out in full the "MT" which I assume stands for "main text". Also below eq (C.1) I would suggest to include a pointer to the definition of $F$ in (A.1), so that each appendix can be read independently.

Recommendation

Ask for minor revision

  • validity: -
  • significance: -
  • originality: -
  • clarity: -
  • formatting: -
  • grammar: -

Author:  Gianmarco Perrupato  on 2024-11-15  [id 4963]

(in reply to Report 2 on 2024-11-01)
Category:
answer to question

1) In order to make the introduction more complete, we decided to incorporate Section 2.1 into it, and we added some comments about the references. In the new version of the manuscript we discuss the derivation of the dynamical equation in section 2, we discuss the variations of the Fredrickson-Andersen model (FAM) in section 3, and we present the connection between persistence and correlation in section 4. The key difference with respect to previous works is that, while previous studies on the dynamics of the FAM are primarily numerical, in this work we derive analytically the equation of motion for the order parameter on the Bethe lattice.

2) Following the suggestion, we have added a discussion about the results found in Phys. Rev. E 110, 044312 (2024) in the introduction. Additionally, we have expanded section 2.4, which now is section 4, to provide a more detailed examination of relevant results from Phys. Rev. E 110, 044312 (2024), building on our previous mention.

3) In the new version of the manuscript we only use negative/positive.

4) These definitions are original of this work. We added a comment to clarify this point at the beginning of the section

5) In order to make the discussion more clear, we summarized in more detail between equations (21) and (22) some of the results of Phys Rev E 110, 044312 (2024). Below Tc the Markov chain defined by the dynamics of the system reduces into separated components. We identify each component with an equilibrium state. Each equilibrium state is in correspondence with an extensive cluster of spins that are permanently blocked. A crucial point is that the “soft” spins, namely those that are not permanently blocked, become non-trivially correlated inside an equilibrium state. This is a consequence of the presence of permanently blocked spins. We added an example of correlations induced by blocked spins in Figure 7 of the new draft. Due to these non-trivial correlations, the magnetization mi of a soft spin si inside an equilibrium state is in general different from the average magnetization 1 − 2p computed with the factorized measure.

6) In the new version we extended the conclusions, including a discussion about the relation between statics and dynamics. The main point is that the equation we derive (see Eq. (2)) holds in the Fredrickson-Andersen model even if the states are not critical, implying that in general criticality in the statics is not a necessary condition for criticality in the dynamics. We are currently investigating the underlying physical reasons behind this scenario, and its possible occurrence in other models.

7) As suggested by the referee we changed MT with main text, and we added a pointer to the definition of F.

---

## Round 2 · List of Changes

1) In order to make the introduction more complete, we decided to incorporate Section 2.1 into it. In the new version of the manuscript we discuss the derivation of the dynamical equation in section 2, we discuss the variations of the Fredrickson-Andersen model (FAM) in section 3, and we present the connection between persistence and correlation in section 4

2) we extended the discussion about Phys Rev E 110, 044312 (2024) in section 4. In particular we added a new figure (figure 7) to give an example of correlations induced by the presence of permanently blocked spins

3) We extended the conclusions with a comment regarding the connection between statics and dynamics

---

## Editorial Decision

published